# Fibrinogen αC-subregions critically contribute blood clot fibre growth, mechanical stability, and resistance to fibrinolysis

Helen R McPherson[1], Cedric Duval[1], Stephen R Baker[2], Matthew S Hindle[1], Lih T Cheah[1], Nathan L Asquith[3], Marco M Domingues[4], Victoria C Ridger[5], Simon DA Connell[6], Khalid M Naseem[1], Helen Philippou[1], Ramzi A Ajjan[1], Robert AS Ariëns[1]*

[1]Discovery and Translational Science Department, Leeds Institute of Cardiovascular and Metabolic Medicine, University of Leeds, Leeds, United Kingdom; [2]Department of Physics, Wake Forest University, Winston Salem, United States; [3]Division of Hematology, Brigham and Women's Hospital, Harvard Medical School, Boston, United States; [4]Instituto de Medicina Molecular - João Lobo Antunes, Faculdade de Medicina, Universidade de Lisboa, Lisbon, Portugal; [5]Department of Infection, Immunity and Cardiovascular Disease, University of Sheffield, Sheffield, United Kingdom; [6]Molecular and Nanoscale Physics Group, University of Leeds, Leeds, United Kingdom

*For correspondence:
R.A.S.Ariens@leeds.ac.uk

Competing interest: The authors declare that no competing interests exist.

**Abstract** Fibrinogen is essential for blood coagulation. The C-terminus of the fibrinogen α-chain (αC-region) is composed of an αC-domain and αC-connector. Two recombinant fibrinogen variants (α390 and α220) were produced to investigate the role of subregions in modulating clot stability and resistance to lysis. The α390 variant, truncated before the αC-domain, produced clots with a denser structure and thinner fibres. In contrast, the α220 variant, truncated at the start of the αC-connector, produced clots that were porous with short, stunted fibres and visible fibre ends. These clots were mechanically weak and susceptible to lysis. Our data demonstrate differential effects for the αC-subregions in fibrin polymerisation, clot mechanical strength, and fibrinolytic susceptibility. Furthermore, we demonstrate that the αC-subregions are key for promoting longitudinal fibre growth. Together, these findings highlight critical functions of the αC-subregions in relation to clot structure and stability, with future implications for development of novel therapeutics for thrombosis.

## Introduction

Fibrinogen is a major component of the blood, normally circulating at concentrations between 1.5 and 4 mg/mL (*Bridge et al., 2017*). The fibrinogen molecule is composed of two pairs of three chains each ($A\alpha_2B\beta_2\gamma_2$). The N-termini of these chains form the central E-region, with the Aα- and Bβ-chain N-termini forming the fibrinopeptides A and B (FpA and FpB), respectively. The E-region is connected to two distal D-regions by two coiled-coil regions comprising all three chains. While the C-termini of the Bβ- and γ-chains end in the D-region, the Aα-chains extend for a substantial length from the D-region (*Undas and Ariëns, 2011*). This extension is known as the αC-region and is composed of two main subsections, a highly flexible αC-connector (α221-α391) and a globular domain known as the αC-domain (α392-α610) (*Tsurupa et al., 2002*).

The majority of the fibrinogen molecule has been resolved by crystallography with the exception of the N-termini of all three chains, and the C-terminus of the Bβ- (residues 459–461) and γ-chain (residues 395–411) (*Weisel and Litvinov, 2017*). The αC-region is the largest unresolved section, which is likely due to its intrinsically disordered nature (*Spraggon et al., 1997*; *Kollman et al., 2009*). This region has previously been characterised in some detail by nuclear magnetic resonance, atomic force microscopy (AFM), and transmission electron microscopy (*Veklich et al., 1993*; *Burton et al., 2006*; *Protopopova et al., 2015*; *Protopopova et al., 2017*). The αC-connector is the least structured of this region and connects the coiled-coil region to an αC-domain (*Weisel and Medved, 2001*; *Tsurupa et al., 2002*). The αC-domain forms a compact structure with two subdomains Aα392–503 and Aα504–610, with Aα392–503 containing a series of β-sheets (*Tsurupa et al., 2009*). In its native structure, the αC-domain of fibrinogen has been shown to associate with the central E-region through electrostatic interactions and is released by thrombin through FpB cleavage (*Veklich et al., 1993*; *Litvinov et al., 2007*).

Fibrinogen is converted to fibrin by thrombin to form a fibrin fibre network, which incorporates red blood cells (RBCs) and platelets to produce a clot that prevents blood loss following vascular injury. Thrombin first cleaves FpA, allowing for the formation of fibrin oligomers, and protofibrils (*Weisel and Litvinov, 2017*). FpB is cleaved by thrombin at a slightly slower rate, facilitating the release of the αC-region and resulting in interactions between adjacent αC-regions to promote lateral protofibril aggregation and fibre diameter growth (*Cilia La Corte et al., 2011*; *Weisel and Litvinov, 2017*). The clot is further stabilised by activated factor XIII (FXIII)-mediated cross-linking of the α- and γ-chains (*Pisano et al., 1968*).

Previous investigations into the function(s) of the αC-region have been based on recombinant fibrinogen (α251), proteolytic digestion products of fibrinogen (fragment X), and dysfibrinogenemia variants, including Marburg, Lincoln, Mahdia, Milano III, and Otago (*Koopman et al., 1992*; *Gorkun et al., 2002*; *Furlan et al., 1994*; *Ridgway et al., 1996*; *Ridgway et al., 1997*; *Gorkun et al., 1998*; *Collet et al., 2005*; *Amri et al., 2017*). In the case of dysfibrinogenemia, data can be challenging to interpret due to a number of factors such as patients not always being homozygous, the presence of a combination of fibrinogen species in the plasma, insertion of additional residues into the α-chain, and/or low circulating levels of fibrinogen, in addition to heterogeneity in post-translational fibrinogen modifications between individuals (*Koopman et al., 1992*; *Ridgway et al., 1996*; *Ridgway et al., 1997*; *Amri et al., 2017*). Investigations with proteolytic fragments, dysfibrinogenemia samples, and recombinant fibrinogen have provided some insights into the possible roles of the fibrinogen αC-region. These include effects on fibrinolysis rates, lateral aggregation, and fibre thickness, in addition to effects on FXIII cross-linking and the mechanical stability of the clots (*Gorkun et al., 2002*; *Gorkun et al., 1998*; *Collet et al., 2005*; *Tsurupa et al., 2009*; *Helms et al., 2012*). However, the respective roles of each of the two sections of the αC-region (the connector and the αC-domain) are hitherto unexplored.

We hypothesise that the connector and the αC-domain have differential effects on fibrin clot network properties, and our work explores the functional role of these two key subregions of the αC-region by producing two recombinant α220 and α390 fibrinogen variants in a mammalian expression system. Fibrinogen α390 was truncated before the start of the αC-domain and therefore lacks the C-terminal domain, while α220 was truncated at the start of the α-connector, resulting in the removal of the entire αC-region. Through the use of these truncated fibrinogens, we aimed to gain a greater understanding of how each αC-subregion influences blood clot structure and function.

## Results

### Impact of αC-subregions on clot structure

The structural differences between WT fibrinogen and the two truncated variants are shown by schematic images (*Figure 1A*). The α390 variant is truncated just before the start of αC-domain, whereas the α220 variant is truncated at the start of the αC-connector and therefore lacks both subregions. The nature of the fibrinogen variants with deletions to the αC-subregions was confirmed by native PAGE and reducing SDS-PAGE (*Figure 1B and C*). Both fibrinogen truncations migrated further compared to WT, with α220 electrophoresing the furthest (*Figure 1B*), in agreement with their reduced molecular weight. One band was observed for each sample, indicating homogeneous species. The WT showed

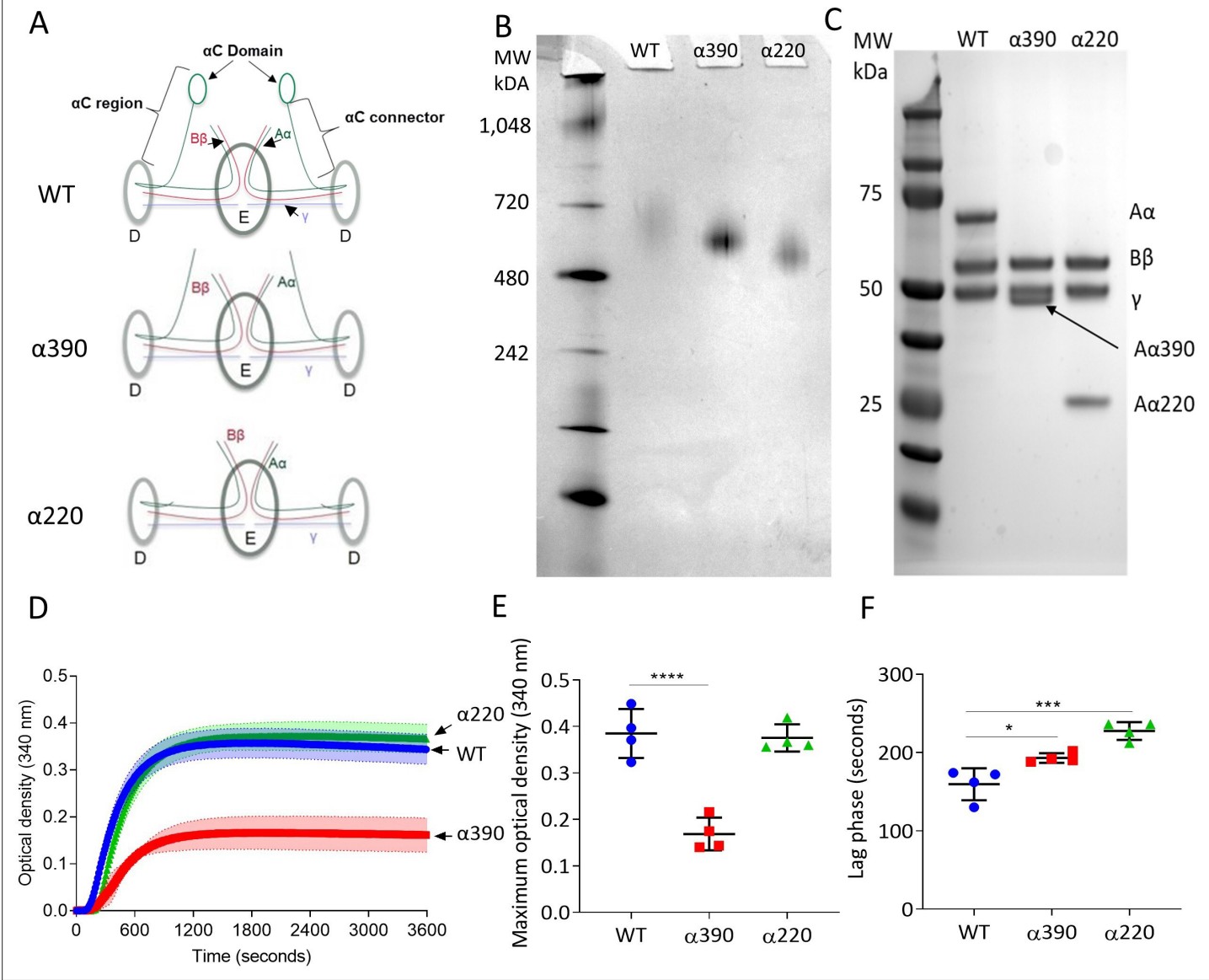

**Figure 1.** Initial characterisation of the recombinant truncated fibrinogen. (**A**) Schematic of WT, α390, and α220 highlighting the differences in structure of αC-region. (**B**) Native PAGE gel of the recombinant fibrinogens showed increased migration with decreasing protein size, and a single band for each fibrinogen α-chain variant and WT. (**C**) Reducing SDS-PAGE gel of the truncated fibrinogens, showing intact β- and γ-chains, and the correct size for the α-chains. All preparations showed no degradation and high purity. The arrow indicates the α-chain for α390 just underneath the γ-chain. (**D**) Turbidity curve of the truncated and WT fibrinogens showing an increase in optical density over time. (**E**) The maximum optical density was significantly reduced for α390, but not α220, compared to WT. (**F**) The lag phase was significantly increased for α390, then α220, compared to WT. Results shown as mean ± SD, n = 4, *p<0.05, ***p<0.001, ****p<0.0001 by one-way ANOVA with Dunnett's multiple comparison test relative to WT.

The online version of this article includes the following figure supplement(s) for figure 1:

**Source data 1.** Raw and uncropped gels for (*Figure 1A*) and (*Figure 1B*) and turbidimetric assay data (*Figure 1D,F*).

**Figure supplement 1.** NativePAGE gel-recombinant WT (rWT) and plasma-purified WT (pWT).

no difference in migration compared with plasma-purified fibrinogen (*Figure 1—figure supplement 1*). Reducing SDS-PAGE confirmed that the lower molecular weight was due to truncation of the fibrinogen α-chain (*Figure 1C*), whilst both β- and γ-chains migrated to the same point as the respective chains of WT fibrinogen. The reduced molecular weight of the α-chains was 42 kDa for α390 and 25 kDa for α220, as predicted, with the calculated molecular weights for the hexameric $(A\alpha_2 B\beta_2 \gamma_2)$-truncated fibrinogens being 290 kDa and 256 kDa, respectively.

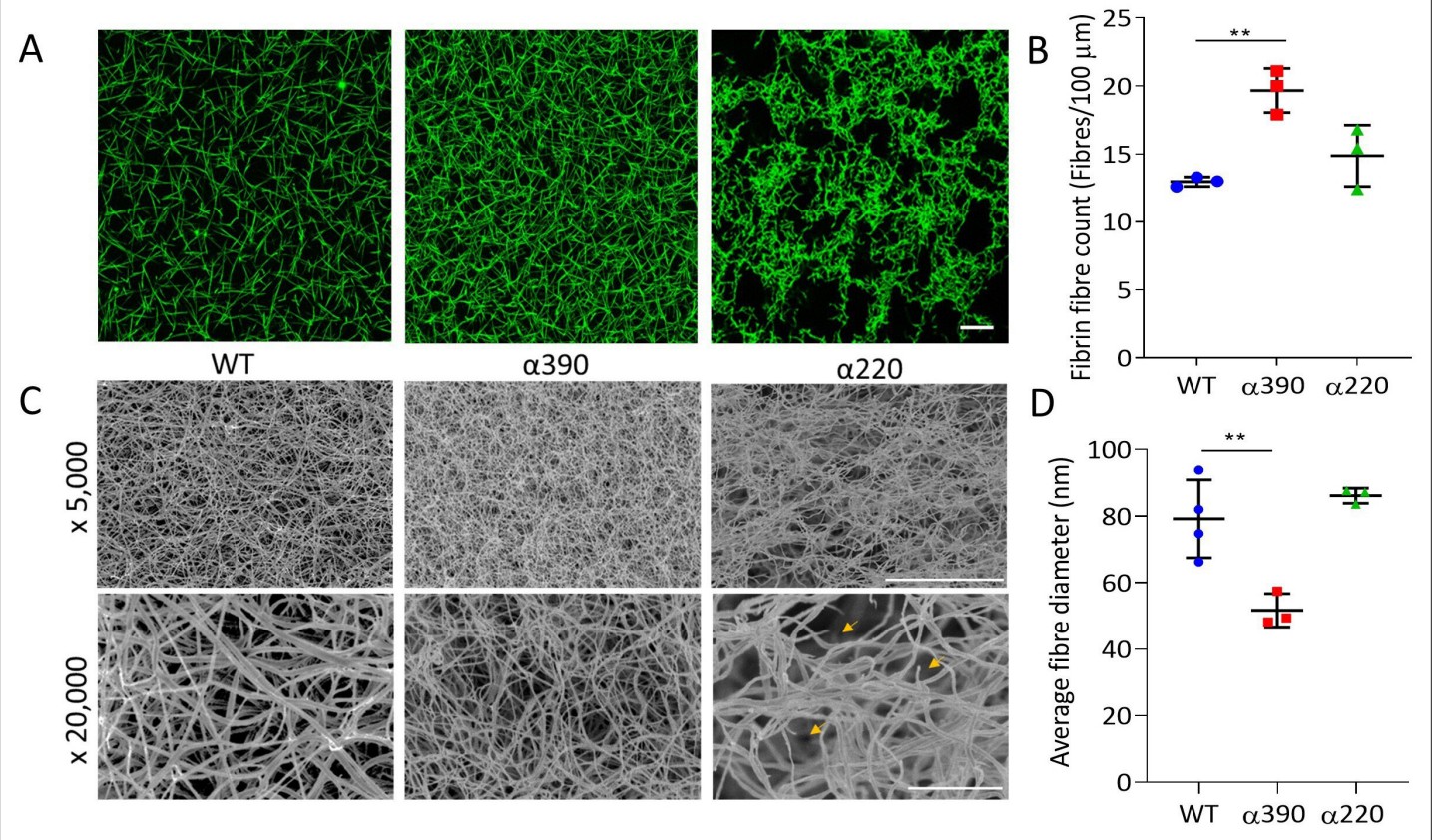

**Figure 2.** Fibrin structural changes observed with truncation to the fibrinogen α-chain. (**A**) Representative laser confocal microscopy images of clots composed of truncated and WT fibrinogens, scale bar is 20 μm. (**B**) Clot fibrin fibre density was significantly increased for α390, but not α220, compared to WT. (**C**) Representative scanning electron microscopy images of clots composed of either truncated fibrinogen or WT, scale bar is 10 μm for ×5000 and 2 μm for the ×20,000 magnifications. Yellow arrows on the ×20,000 magnification images for α220 highlight the numerous fibre ends visible throughout the clot. (**D**) Average fibre diameter calculated from the ×20,000 magnification images showed a significantly decreased diameter for α390, but not α220, compared to WT. Laser scanning confocal images were taken on an LSM880 inverted laser scanning confocal microscope (Zeiss; Cambridge, UK) using a ×40 magnification oil objective. Scanning electron microscope images were taken on an SU8230 Ultra-High-Resolution Scanning Electron Microscope (Hitachi, Tokyo, Japan). Results shown as mean ± SD, n = 3 confocal and scanning electron microscopy n = 3 (α390 and α220) and n = 4 WT, **p<0.01 by one-way ANOVA with Dunnett's multiple comparison test relative to WT.

The online version of this article includes the following source data for figure 2:

**Source data 1.** Fibre count (**Figure 2B**) and diameter (**Figure 2D**).

Fibrin polymerisation by turbidimetry was similar for clots produced with WT (0.385 ± 0.052 OD) and α220 (0.376 ± 0.029 OD, p=0.9228) fibrinogens (**Figure 1D and E**). In contrast, clots produced with α390 (0.169 ± 0.035 OD, p<0.0001) fibrinogen showed significantly reduced maximum optical density (**Figure 1E**), indicating the formation of a fibrin clot with thinner fibres and a denser structure. There was a step-wise increase in the lag phase from WT (159.5 ± 20.3 s) to α390 (193 ± 6.2, p=0.0143) and α220 (227.5 ± 11.4, p=0.0001) (**Figure 1F**).

Fibrin network structure was visualised using laser scanning confocal microscopy (LSCM), which showed distinct clot architectures for α390 and α220 fibrin compared with WT and each other (**Figure 2A**). In agreement with turbidimetric analysis, a denser clot structure was observed for clots made with α390 fibrinogen compared to WT. However, α220 clots were much more heterogeneous and their structure showed clumps of regions with highly branched fibres of stunted length compared to WT, leaving very large pores that were observed throughout. No difference in average fibre count was observed between WT (12.97 ± 0.35 fibre/100 μm) and α220 (14.87 ± 2.25 fibre/100 μm, p=0.3212) (**Figure 2B**), but the fibre count was significantly increased for α390 clots (19.6 ± 1.79 fibre/100 μm, p=0.0041). Despite the clear difference in clot structure between α220 and WT, the

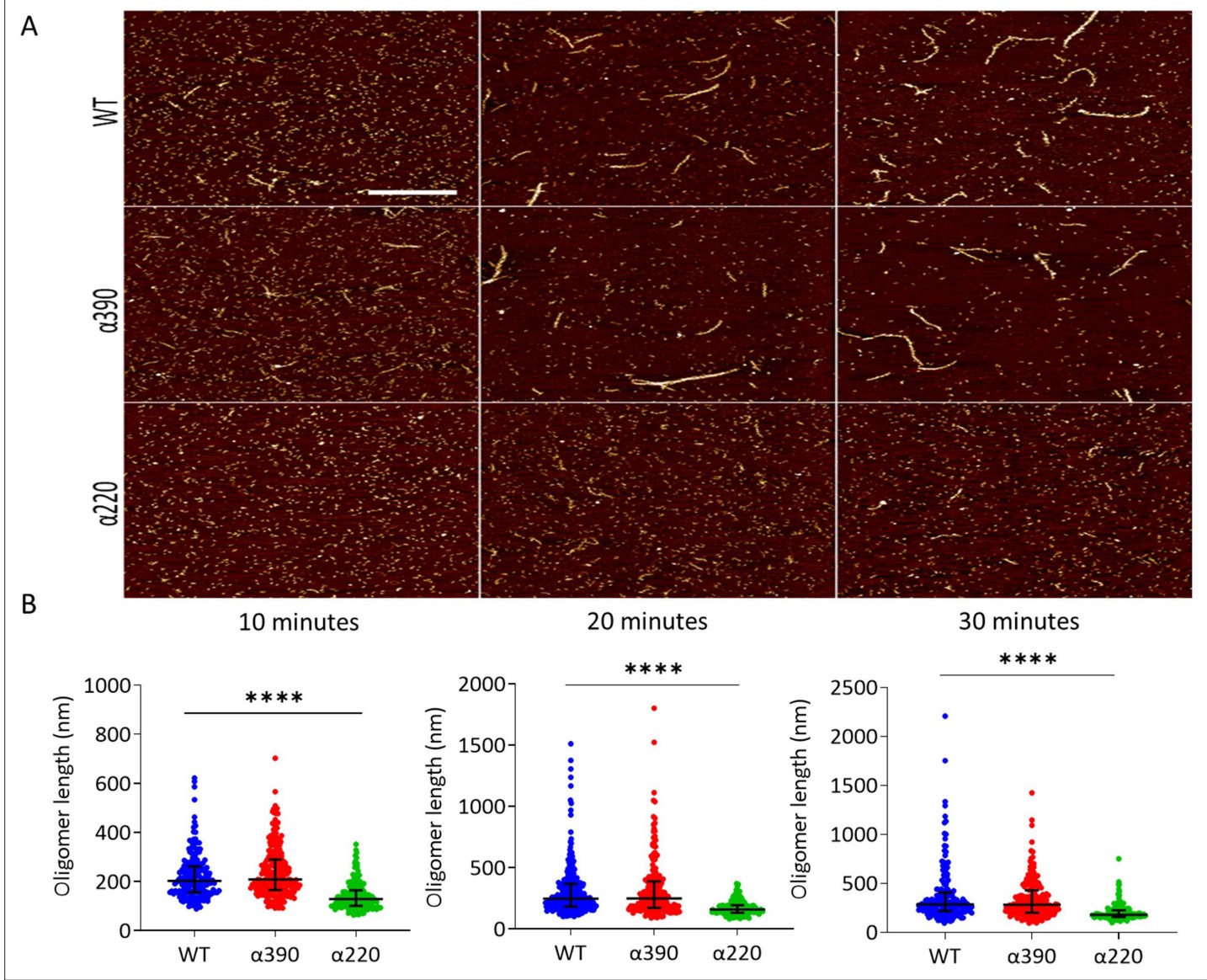

**Figure 3.** Early fibrin polymers length comparison. (**A**) Representative images of fibrin polymers lengths for truncated and WT fibrinogens at 10, 20, and 30 min, scale bar is 1 µm. (**B**) Average polymer length was significantly decreased for α220, but not α390, at all time points compared to WT. Each point in (**B**) represents a single measured polymer; at least 165 polymers were measured for each time point/fibrinogen variant, and at least three replicates were made of each sample. Results shown as median ± IQR, ****p<0.0001 by Kruskal–Wallis test with Dunn's multiple comparison test relative to WT.

The online version of this article includes the following source data for figure 3:

**Source data 1.** Oligomer lengths (*Figure 3B*).

average fibre count was similar, suggesting that α220 truncation affects clot organisation and fibre growth, but not the overall number of fibres that are initiated within the clot.

Subsequently, fibrin clot ultrastructure was investigated using SEM (*Figure 2C*). Consistent with LSCM and turbidimetric analysis, network structure for α390 was denser compared with WT clots. Compared to α390, clots made with α220 and WT fibrinogen had similar network densities, but numerous fibre ends were visible in α220 clots, which were not observed in either of the other two clots (*Figure 2C*, yellow arrows). Average fibre diameter was reduced for α390 (51.67 ± 5.03 nm, p=0.0059) compared to WT (79.19 ± 11.72 nm), but not for α220 (86.15 ± 2.26 nm, p=0.4801) (*Figure 2D*).

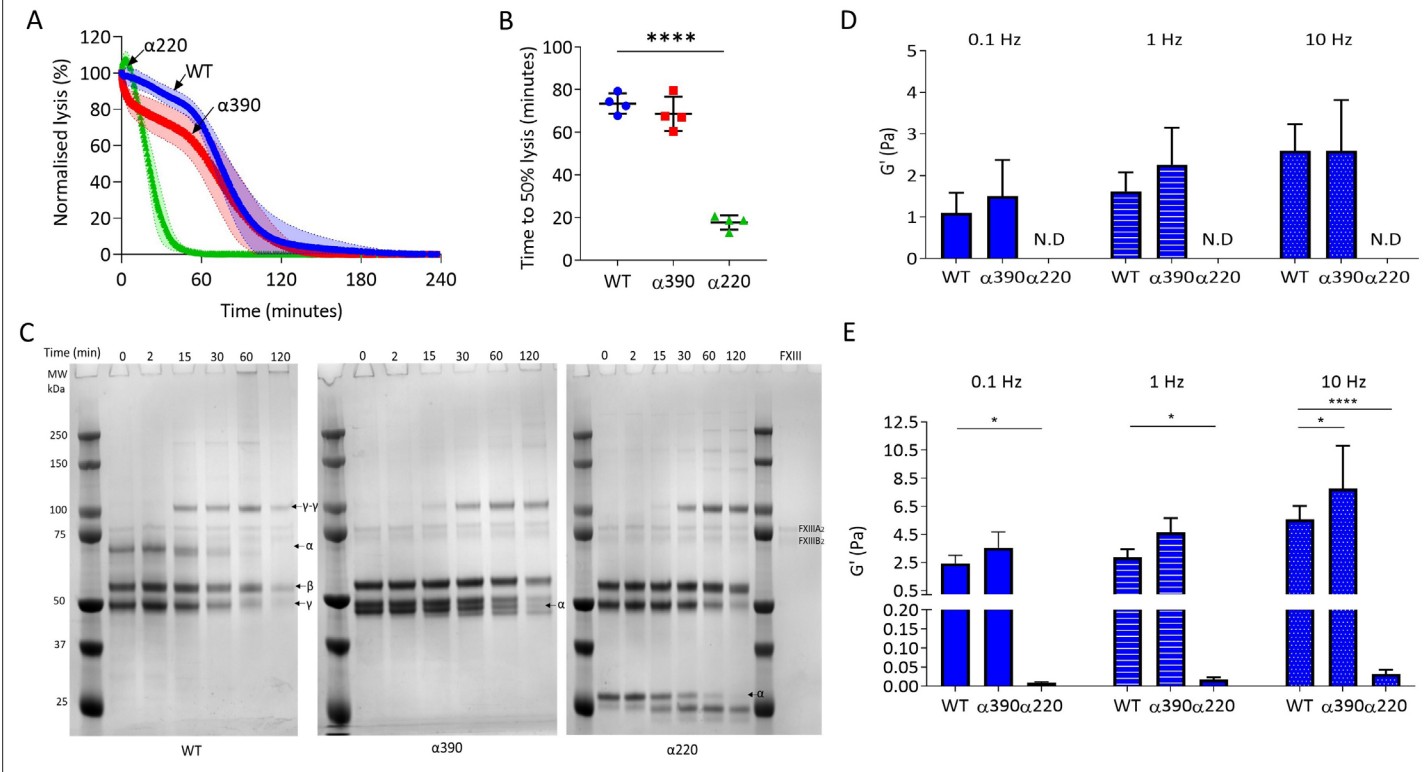

**Figure 4.** αC-connector provides clot stability. (**A**) Normalised fibrinolysis curve over time of truncated and WT fibrinogens. (**B**) Time to 50% lysis was significantly decreased for α220, but not α390, compared to WT. (**C**) Representative SDS-PAGE gels of truncated and WT fibrinogens undergoing cross-linking by FXIII, showing α- and γ-chain cross-linking in the variants. (**D**) Elastic modulus (G') at 0.1 Hz, 1 Hz, and 10 Hz, in the absence of FXIII, showed that clot stiffness was not different between α390 and WT, but could not be determined (N.D) for α220. (**E**) Elastic modulus (G') at 0.1 Hz, 1 Hz, and 10 Hz, in the presence of FXIII, showed that α390 markedly increased, whilst α220 significantly decreased, clot stiffness compared to WT. n = 4. Results shown as mean ± SD, n = 4 for (**A**), (**B**), (**D**), and (**E**) n = 4 for WT and α390 and n = 3 in (**E**) for α220, *p<0.05, **p<0.01, ***p<0.001, ****p<0.0001 by one-way ANOVA with Dunnett's multiple comparison test (**B**) and two-way ANOVA with Dunnett's multiple comparison test graphs (**D**) relative to WT.

The online version of this article includes the following source data, source code, and figure supplement(s) for figure 4:

**Source code 1.** Code used in MATLAB for fibrin microrheology analysis.

**Source data 1.** Fibrinolysis data (*Figure 4A and B*), raw and uncropped gels for (*Figure 4C*), and fibrin microrheology data for *Figure 4D and E*.

**Figure supplement 1.** Fibrinolysis by laser scanning confocal microscopy.

**Figure supplement 1—source data 1.** Fibrinolysis times (*Figure 4—figure supplement 1*).

## Protofibril growth

AFM was used to study the role of the αC-subregions during early polymerisation steps. *Figure 3A* shows representative early polymerisation images of the fibrinogen variants over time under diluted in vitro conditions to slow the reaction down from seconds to minutes. An increase in polymer length was observed for both WT and α390, whereas α220 showed limited polymer growth (*Figure 3B*). At 10 min, similar oligomer lengths were observed for WT (223 ± 94 nm) and α390 (240 ± 103 nm, p=0.3596), whereas α220 average length was shorter at 142 ± 56 nm (*P* < 0.0001). A similar pattern of reduced oligomer growth for α220 was observed at 20 min WT (319 ± 263 nm), α390 (326 ± 245 nm *P* => 0.9999) and α220 (173 ± 58 nm p < 0.0001) and 30 min WT (372 ± 284 nm), α390 (341 ± 202 nm *P* => 0.9999) and α220 (204 ± 84 nm p < 0.0001).

## Clot mechanics and in vitro fibrinolysis

Next, the effect of αC-subregions on clot resistance to lysis and viscoelastic properties was investigated. Initial investigations into lysis by LSCM showed there was no defined lysis front for α220 clots, likely due to the highly porous structure (*Figure 4—figure supplement 1A*) and there was no difference in time to 50% lysis between WT and α390 clots (*Figure 4—figure supplement 1B*).

To overcome the lack of a defined lysis front, lysis was studied by turbidity after layering fibrinolytic factors on preformed clots (*Figure 4A*). Time to 50% lysis was similar for α390 and WT clots (68.6 ± 8.0 vs. 73.4 ± 4.8 min, p=0.4138), whereas α220 clot lysis was considerably faster (17.7 ± 3.3 min, p<0.0001) (*Figure 4B*).

The αC-domain contains lysines which are cross-linked by FXIII to γ-glutamyl-ε-lysyl bonds with glutamines in the αC-connector of adjacent fibrin molecules, some of which are still present in α390 (Q221 [and/or 223], Q237, Q328, and Q366) (*Mouapi et al., 2016*; *Schmitt et al., 2019*). The entire αC-region is removed in α220, and therefore, all residues involved in cross-linking are absent. Cross-linking analysis by SDS-PAGE showed reductions in α-chain molecular weight over time due to the FpA cleavage by thrombin, noticeable in all fibrinogen variants, particularly after 15 min (*Figure 4C*). Both truncations displayed normal fibrin γ-chain cross-linking and formation of γ-γ dimers, but delayed α-chain cross-linking with both truncations showing un-cross-linked α-chain after 2 hr, while all α-chain monomer had been converted to polymer in WT. Some α-chain cross-linking may have still occurred even in α220 which lacks all known cross-linking residues, suggesting the presence of additional cross-linking sites.

Next, effects of αC-subregions on clot viscoelastic properties were investigated by magnetic tweezers. Without FXIII, α390 clots showed similar storage modulus (G′) at 0.1, 1, and 10 Hz (*Figure 4D*) as WT. G′ is a measure of elastic energy stored through deformation and is linked to clot stiffness. No viscoelastic data could be generated for α220 clots without FXIII due to highly porous nature of these clots and inability of superparamagnetic beads to remain trapped by the weak fibrin network. With FXIII, the G′ was higher for both WT and α390 than without FXIII (*Figure 4D and E*). Cross-linked α220 clots showed markedly reduced G′ over all frequencies compared to WT, whereas α390 clots showed higher elastic modulus compared to WT with FXIII at 10 Hz (*Figure 4E*). In experiments with and without FXIII, there was a trend for α390 having a higher elastic modulus compared to WT. This observation is likely due to increased fibre branching and clot density in this variant.

## Clot contraction

To explore the role of αC-subregions in mediating interactions with blood cells, whole blood from F*ga*<sup>-/-</sup> mice supplemented with variant or WT fibrinogen was analysed for clot contraction, which occurs secondary to platelet contractile forces pulling fibrin fibres while trapping RBCs. Clots formed and contracted normally for WT and α390, but no visible clot was formed for α220 (*Figure 5A*), and as a result neither clot contraction nor weight could be analysed. No difference was observed in clot retraction kinetics (*Figure 5B*) or clot weight between WT and α390 (*Figure 5C*).

Comparing the supernatant (activated) of contracted clots to corresponding non-activated samples indicated that platelets were fully incorporated into the contracted clot (*Figure 5—figure supplement 1A and B*). Despite the lack of distinct clot formation, α220 showed reduced number of platelets in the supernatant, likely due to normal platelet activation and formation of platelet aggregates, which agrees with an increased FFC/SCC platelet profile in this sample. Control samples containing tissue factor, but no fibrinogen, also showed reduced platelet numbers in the supernatant. Control samples without tissue factor and fibrinogen showed similar platelet numbers for non-activated and activated as no clot formed and platelets were not activated. Examination of the supernatant of contracted clots to non-activated samples for RBC retention showed similar retention between WT and α390, while α220 clots demonstrated reduced RBC retention compared with WT (*Figure 5D*). The amount of RBC retention in α220 was similar to the activated control without fibrinogen, suggesting that platelet aggregates may be trapping some RBCs.

Next, we investigated fibrin interaction with platelets by flow cytometry. Upon ADP and PAR4 activation, fibrinogen-positive platelets increased to the same extent for WT, α390, and α220, but not fibrinogen γ′, consistent with its lack of integrin-binding C-terminal AGDV motif (*Figure 5—figure supplement 2A*). However, fibrinogen median fluorescence intensity (MFI) for α220 was decreased, suggesting reduced fibrinogen bound per platelet. In contrast, α390 displayed similar binding levels as WT (*Figure 5—figure supplement 2B*). Despite minor differences in binding, these data indicate that clot contraction differences are not driven by a complete inability of α220 to bind platelets.

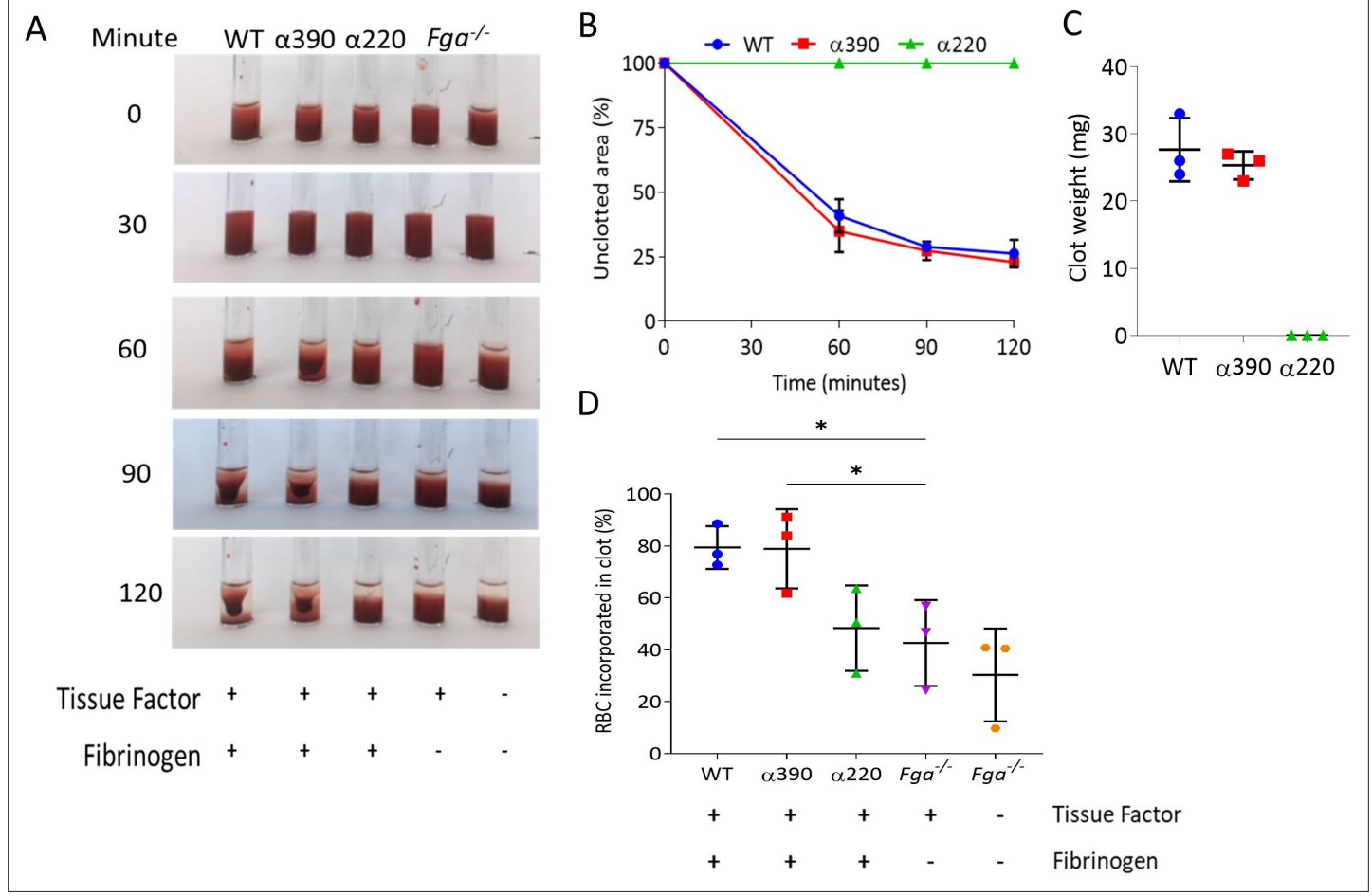

**Figure 5.** Whole blood clot contraction. Whole blood clots were prepared from whole blood from *Fga*[-/-] non-activated mice supplemented with either truncated fibrinogen or WT, no fibrinogen (controls). (**A**) Representative images of whole blood clot formation and contraction over time, showing the formation of a defined clot for WT and α390, but no visible clotting for α220 and controls. (**B**) Whole blood clot contraction kinetics was similar between α390 and WT, but not quantifiable for α220. (**C**) Final clot weight was not different between α390 and WT, but not quantifiable for α220. (**D**) The percentage of red blood cells incorporated in the contracted clot was similar between α390 and WT, whilst that of α220 was similar to the control containing tissue factor but no fibrinogen. Results shown as mean ± SD, n = 3, *p<0.05 by one-way ANOVA with Dunnett's multiple comparison test relative to the tissue factor-activated control.

The online version of this article includes the following source data and figure supplement(s) for figure 5:

**Source data 1.** Retraction data (*Figure 5B*), weights (*Figure 5C*), and absorbances for red blood cell incorporation (*Figure 5D*).

**Figure supplement 1.** Clot incorporation of platelets.

**Figure supplement 2.** Recombinant fibrinogen is able to bind to platelets.

## Whole blood clot lysis

Thromboelastography was used to study clot formation, firmness, and fibrinolysis in whole blood from *Fga*[-/-] supplemented with fibrinogen variants. EXTEM (tissue factor as trigger) showed no clot formation in whole blood from the *Fga*[-/-] mice, When supplemented with WT, clot formation occurs and shows a similar profile to whole blood from a C57BL/6 mice (*Figure 6—figure supplement 1*). EXTEM showed extended clotting time for α390 (110.7 ± 18.5 s, p=0.7590) and more markedly α220 (425.7 ± 111.4 s, p=0.0014) compared to WT (65.3 ± 16.8 s) (*Figure 6A and B*), consistent with the prolonged lag-phase observed in turbidity. Clot firmness was reduced for both truncations compared to WT (45.7 ± 6.3 mm), with α220 (7.0 ± 2.0 mm, p=0.0005) being the weakest and α390 displaying an intermediate phenotype (24 ± 8.5 mm, p=0.0099) (*Figure 6C*). The clot firmness trace for α220 returned to baseline, with a similar but not complete decline for α390 (*Figure 6A*).

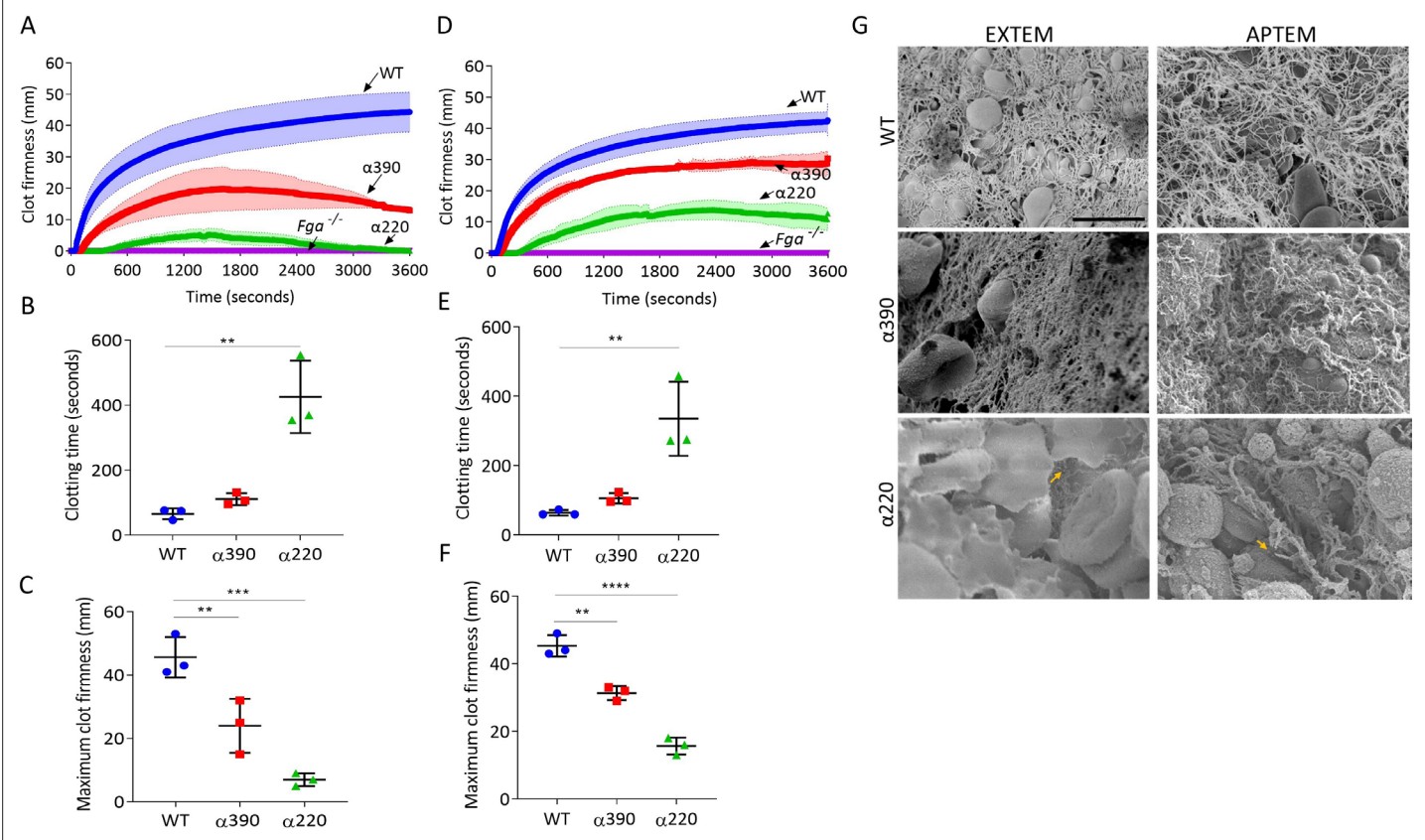

**Figure 6.** Fibrinolysis is the crucial factor for clot instability. Whole blood clots were prepared with whole blood from *Fga*⁻/⁻ mice supplemented with either truncated or WT fibrinogens. (**A**) Whole blood clot formation curves from ROTEM, with clotting activated by tissue factor (EXTEM), showing clot firmness over time. (**B**) Clotting time was significantly increased for α220, but not α390, compared to WT. (**C**) Maximum clot firmness was significantly decreased for α390, and even further for α220, compared to WT. (**D**) Whole blood clot formation curves from ROTEM, with clotting activated by tissue factor and aprotinin to inhibit fibrinolysis (APTEM), showing clot firmness over time. (**E**) Clotting time was significantly increased for α220, but not α390, compared to WT. (**F**) Maximum clot firmness was significantly decreased for α390, and even further for α220, compared to WT. (**G**) Representative scanning electron microscopy images of clots collected and fixed after ROTEM analysis. Images taken on a SU8230 Ultra-High-Resolution Scanning Electron Microscope (Hitachi, Tokyo, Japan). Yellow arrow indicates fibrin, scale bar 5 µm. Results shown as mean ± SD, n = 3, *p<0.05, **p<0.01, ***p<0.001, ****p<0.0001 by one-way ANOVA with Dunnett's multiple comparison test (**B, C, F**) and Kruskal–Wallis test with Dunn's multiple comparison test (**E**) relative to the WT.

The online version of this article includes the following source data and figure supplement(s) for figure 6:

**Source data 1.** ROTEM firmness values (***Figure 6A and D***) and values generated by ROTEM for ***Figure 6B, C, E, and F***.

**Figure supplement 1.** Comparison of ROTEM (EXTEM) profiles of C57BL/6 (*Fga*⁺/⁺), *Fga*⁻/⁻, and *Fga*⁻/⁻ supplemented with 0.5 mg/mL recombinant WT (rWT).

ROTEM clots were then collected and prepared for SEM (***Figure 6G***). WT and α390 clots were composed of fibrin, platelets, and RBCs, while α220 clots showed limited fibrin and were mainly composed of RBCs and platelets. As limited fibrin was observed in α220 clots, the role of fibrinolysis in clot destabilisation was investigated using APTEM (includes aprotinin, an inhibitor of plasmin; ***Figure 6D***). Similar to EXTEM, clotting time was extended (***Figure 6E***) for α390 (105.7 ± 15.0 s, p=0.3558) and α220 (335.0 ± 106.5 s, p=0.0141) compared to WT (63.7 ± 8.1 s). Clot firmness was higher in APTEM than in EXTEM for the truncations (α390, 31.3 ± 2.1 mm, p=0.0012; and α220, 15.7 ± 2.5 mm, p<0.0001), though still reduced compared to WT (45.3 ± 3.2 mm), and both were able to maintain clot firmness over the period of the experiment (***Figure 6D and F***). Fibrin was clearly present for α220 clots after APTEM, and numerous visible fibre ends were again visible (***Figure 6G***). These results indicate that fibrinolysis caused α220 clots to destabilise within 1 hr of clot formation in whole blood.

## Discussion

Our main findings are that without the C-terminal domain (α390), fibrin produces a clot consisting of thinner fibres and composed of a denser structural network with reduced mechanical stability. However, and strikingly, without the whole of the αC-region and deletion of both the C-terminal domain and connector (α220), fibrin longitudinal growth is impaired significantly. Clots produced from α220 fibrinogen have stunted fibres and abnormal network structures that show drastically reduced mechanical and fibrinolytic resistance. These findings demonstrate a far greater role for the fibrinogen αC-region than hitherto thought, with it being critically important for normal fibrin fibre formation, growth, and clot stability.

The loss of the αC-domain in α390 produced a denser clot with thinner fibres compared to WT, as observed by both SEM and LSCM. These findings are reminiscent of a shorter fibrinogen variant α251 (*Collet et al., 2005*). Interestingly, the additional 139 AA residues in α390 did not seem to have affected final clot structure, which was largely similar to fibrin networks made from α251 fibrinogen (*Gorkun et al., 1998*; *Collet et al., 2005*). In contrast, the lack of 31 AA residues in α220, compared with α251, seems to significantly affect fibre growth and clot stability. It thus appears that the αC-connector region, even if only partially present, rescues some of the fibre growth capabilities and the stability of α251 clots, which is greatly impaired when the full connector region is lost in α220.

The transition from longitudinal to lateral growth of protofibrils has been reported to occur at oligomer lengths of 0.6–0.8 µm (*Hantgan et al., 2018*; *Chernysh et al., 2011*). The AFM experiments cannot extend to time points beyond protofibrils growth to 0.6 µm since this also coincides with network gelation. This notwithstanding, at earlier time points, there was comparable growth in protofibril length for WT and α390 fibrin, while α220 had significantly shorter oligomers at every time point. Moreover, LSCM and SEM data showed that the loss of the αC-connector results in short and stunted fibres. Taken together, these results indicate that α220 fibres are unable to support normal longitudinal fibre growth, which would normally lead to a continuous network of fibres without visible fibre ends. Previous data support our conclusions as clots made from fragment $X_2$, where the C-termini of both α-chain end at residue 219, show visible fibre ends (*Gorkun et al., 2002*). Moreover, clots from fragment $X_2$ demonstrated large pores, consistent with our finding with α220 variant, although fibre thickness was different, which may be due to the use of purified bovine fibrinogen or the partial loss of the β-chain during fragment $X_2$ generation (*Gorkun et al., 2002*).

Previously our investigations with a recombinant fibrinogen Double-Detroit with mutations in both 'A' and 'B' knobs resulted in defective knob-hole interactions, therefore preventing protofibril formation, lateral aggregation, and polymerisation (*Duval et al., 2020*). As there was no protofibril initiation and formation, the role of the αC-region in protofibril growth could not be detected. In our current αC-truncations, knob-hole interactions are unchanged, thus allowing protofibril initiation, but longitudinal growth is impaired, altogether indicating a key role for the αC-subregions in protofibril growth, but not initiation.

The change in clot structure and altered cross-linking of fibrin fibres with the α220 variant is likely to contribute to the marked reduction in fibrin network stiffness compared with WT, and the instability of the clots even in the absence of FXIII. In agreement with this, Collet et al. were unable to study permeation for α251 in the absence of FXIII due to clot weakness (*Collet et al., 2005*). Fibrinogen α251 is likely to be mechanically stronger than α220 as there are still at least two glutamine (Q221 [and/or 223] and Q237) and lysine (K208 and K219) residues present that can be cross-linked by FXIII (*Cilia La Corte et al., 2011*). Moreover, our clot contraction data showed a reduction in RBC and platelet retention with α220 clots, while α390 and WT were largely similar. This is consistent with previous data using α251, which demonstrated that α-α cross-linking is essential for the retention of RBC within the clot (*Byrnes et al., 2015*).

The connector region also demonstrated a significant effect on susceptibility of the clot to lysis, which may have been due, at least in part, to altered clot structure, but there are other potential mechanisms. The α-chain has several lysine cleavage sites involved in fibrinolysis, AαK583, AαK230, and AαK206, before lysines in the coiled-coil of all three chains are targeted (*Hudson, 2017*). AαK583 is the most vulnerable and leads to partially degraded fibrinogen within circulation (*Hudson, 2017*). As observed in both turbidimetric and ROTEM experiments for α220, there was a delay in the initial stages of clotting, providing opportunity for fibrinolysis to occur simultaneously with polymerisation. In addition, the conversion of fibrinogen to fibrin exposes t-PA and plasminogen binding sites,

namely, γ312–324 in the γ-nodule and Aα148–160 within the coiled-coil region (*Yakovlev et al., 2000*; *Medved and Nieuwenhuizen, 2017*). These sites are likely already exposed in α220 fibrinogen prior to their conversion to fibrin due to the lack of tethering of the αC-region to the coiled-coil and the E-region, in turn leading to early plasmin generation by t-PA (*Veklich et al., 1993*; *Litvinov et al., 2007*). In support of this notion, α220 clots displayed rapid intrinsic lysis in whole blood when fibrinolysis was not inhibited (EXTEM) and in purified protein data, but not when lysis was inhibited in whole blood (APTEM). This observation was further confirmed by SEM of α220 clots after EXTEM experiments, showing the presence of only minimal amounts of fibrin, whereas fibrin was clearly present throughout the clots after APTEM.

Intrinsic fibrinolysis was also enhanced in clots made from α390, as shown in EXTEM experiments, but to a lesser extent compared with α220. Similarly to α220, however, the effect on fibrinolysis was inhibited in APTEM experiments for α390. To complicate matters, turbidimetric analysis of external fibrinolysis showed no overall difference in lysis of α390 clots compared with WT. This is likely due to the denser clot structure in α390 that prevents movement of plasmin through the clot, thus counterbalancing increased fibrinolysis initiation due to lack of the αC-domain (*Collet et al., 2000*). The denser clot structure observed in the purified system for α390 was less apparent in whole blood, indicating that blood cells prevent the formation of a denser clot. As for the difference in clot lysis comparing α390, α251 and α220, one mechanism may be related to the absence of the $\alpha_2$-antiplasmin cross-linking site Lys303 in both α251 and α220, resulting in reduced incorporation of this antifibrinolytic protein. Non-covalent interactions of $\alpha_2$-antiplasmin have also been reported for the D-region and the C-terminal subdomain of the αC-region, which could further play a role in different lysis patterns of clots made from these two variants (*Tsurupa et al., 2010*). Of note, our data also show an important mechanism for the interplay between clot mechanics and fibrinolysis, with both truncations failing to maintain clot firmness during ROTEM analysis, unlike WT. However, inhibition of fibrinolysis restored maintenance of clot firmness for both truncations, showing that in the absence of the αC-domain and/or αC-connector, fibrinolysis is an important regulator of clot mechanical stability.

Our study had a number of limitations. Due to the low expression levels of the αC-truncation variants, any methodologies that required large amounts of protein, including, for example, in vivo thrombosis studies based on injection of the recombinant variants in *Fga*[-/-] mice, could not be performed. Limitations were also posed by the extreme weakness of the α220 variant, which could not be analysed in some of our experiments without cross-linking by FXIII. Future in vivo studies using mouse models with similar truncations may provide additional information on the role of the αC-region in physiological haemostasis and pathological thrombosis. However, as mentioned above, such models will likely be confounded by the effects of αC-truncations on both protein function and expression level.

In conclusion, our data show that the αC-domain and αC-connector play crucial and distinct roles in fibrin formation, fibre growth, clot structure, and clot stability. While the αC-domain is central to lateral aggregation, fibre thickness, and clot network density, the αC-connector, together with the αC-domain, is key for longitudinal fibre growth, continuity of fibrin fibres, and both mechanical and fibrinolytic clot stability. These novel functions of the αC-connector region enhance our understanding of the role of this fibrinogen region in clot formation, structure, and resistance to lysis. Moreover, our data have important ramifications for potential future therapies, targeting the αC-region of fibrinogen for the reduction of thrombosis risk.

## Materials and methods

### Key resources table

| Reagent type (species) or resource | Designation | Source or reference | Identifiers | Additional information |
|---|---|---|---|---|
| Genetic reagent (*Mus musculus*) | *Fga*[-/-] | Kind gift from Dr Jay Degen, Cincinnati Children's Hospital Medical Centre | PMID:7649481 | |

*Continued on next page*

*Continued*

| Reagent type (species) or resource | Designation | Source or reference | Identifiers | Additional information |
|---|---|---|---|---|
| Genetic reagent (*M. musculus*) | *Fga*⁺/⁻ | Kind gift from Dr Jay Degen, Cincinnati Children's Hospital Medical Centre | | PMID:7649481 |
| Cell line (*Cricetulus griseus*) | CHO-αβγ | Kind gift from Dr Susan Lord, University of North Carolina | Expressing WT human fibrinogen | PMID:8418831 |
| cell line (*C. griseus*) | CHO-βγ | Kind gift from Dr Susan Lord, University of North Carolina | Expressing β- and γ-chain of human fibrinogen | PMID:8418831 |
| Cell line (*C. griseus*) | CHO-αβ | Kind gift from Dr Susan Lord, University of North Carolina | Expressing α- and β-chain of human fibrinogen | PMID:8418831 |
| Transfected construct (human) | pMLPAα | Kind gift from Dr Susan Lord, University of North Carolina | | PMID:8418831 |
| Transfected construct (human) | pMLPAα390 | This paper | | Ariëns Lab |
| Transfected construct (human) | pMLPAα220 | This paper | | Ariëns Lab |
| Biological sample (human) | Human plasminogen-depleted plasma-purified fibrinogen | Merck | 341578 | |
| Biological sample (human) | Glu-plasminogen | Enzyme Research Laboratories | HPG 2001 | |
| Biological sample (human) | FXIII | Zedira | T007 | |
| Biological sample (human) | Tissue factor (PPP reagent) | Stago | 86193 | |
| Biological sample (human) | Thrombin | Merck | 605190 | |
| Antibody | CD41 Alexa Fluor 700 (anti-mouse monoclonal) (MWReg30) | BioLegend | 133925 | (5 µg/mL) |
| Antibody | CD62P PE (anti-mouse monoclonal) (RMO-1) | BioLegend | 148305 | (10 µg/mL) |
| Antibody | CD63 APC (anti-mouse monoclonal) (NVG-2) | BioLegend | 143905 | (5 µg/mL) |
| Antibody | IgG2a PE (monoclonal Mouse) (MOPC-173) | BioLegend | 400211 | (10 µg/mL) |
| Antibody | IgG2a APC (monoclonal Rat) (RTK 2758) | BioLegend | 400511 | (5 µg/mL) |
| Antibody | FITC fibrinogen (anti-human polyclonal) | Agilent Technologies | F0111 | (60 µg/mL) |
| Peptide, recombinant protein (human) | WT | This paper | | Purified from CHO cells, WT fibrinogen, Ariëns Lab |
| Peptide, recombinant protein (human) | α390 | This paper | | Purified from CHO cells, fibrinogen with α-chain truncated at α390, Ariëns Lab |

*Continued on next page*

*Continued*

| Reagent type (species) or resource | Designation | Source or reference | Identifiers | Additional information |
|---|---|---|---|---|
| Peptide, recombinant protein (human) | α220 | This paper | | Purified from CHO cells, fibrinogen with α-chain truncated at α220, Ariëns Lab |
| Peptide, recombinant protein (human) | γ'/γ' | This paper | | Purified from CHO cells, fibrinogen homozygous in γ'-chain, Ariëns Lab |
| Peptide, recombinant protein (human) | t-PA | Pathway Diagnostics | TC41072 | |
| Chemical compound, drug | PAR4 | Cambridge BioScience | ANA60218-5 | |
| Commercial assay or kit | Alexa Fluor 488 using a protein labelling kit | Thermo Fisher Scientific | A10235 | |
| Commercial assay or kit | QuikChange II Site-Directed Mutagenesis Kit | Agilent Technologies | 200524 | |
| Software, algorithm | ImageJ software | ImageJ (http://imagej.nih.gov/ij/) | RRID:SCR_003070 | |
| Software, algorithm | GraphPad Prism | GraphPad Prism (https://graphpad.com) | RRID:SCR_015807 | Version 8 |
| Software, algorithm | MATLAB | MathWorks (http://mathorks.com) | RRID:SCR_001622 | |
| Software, algorithm | OriginLab | OriginLab (http://originlab.com) | RRID:SCR_014212 | |
| Other | Dynabeads M-450 Epoxy | Invitrogen | 14011 | |

## Materials

Human thrombin and human plasminogen-depleted plasma-purified fibrinogen were purchased from Merck (Cramlington, UK) and reconstituted in double-distilled water (ddH$_2$O). Tissue plasminogen activator (t-PA) was from Pathway Diagnostics (Dorking, UK) and human Glu-plasminogen from Enzyme Research Laboratories (Swansea, UK) and were prepared in ddH$_2$O. Human FXIII from Zedira (Darmstadt, Germany) was further purified from contaminating albumin and glucose by Hiload 16/60 superdex 200 (GE Healthcare, Little Chalfont, UK) gel filtration as previously described (*Hethershaw et al., 2014*). All purified proteins were stored at –80°C prior to use. All other chemicals used were obtained from Sigma-Aldrich (Gillingham, UK) unless stated otherwise.

## Experimental animals

All animals were maintained in individually ventilated cages, at 21°C with 50–70% humidity, light/dark cycle 12/12 hr, and on standard diet ad libitum. Fibrinogen knockout (*Fga*)$^{-/-}$ mice (*Suh et al., 1995*), on C57BL/6 background, were generated by crossing male *Fga*$^{-/-}$ with female *Fga*$^{+/-}$ mice, and genotypes were determined using real-time PCR (Transnetyx, Cordova, USA). Mice were bled for whole blood contraction and rotational thromboelastometry experiments. Procedures were performed according to accepted standards of humane animal care, approved by the ethical review committee at the University of Leeds, and conducted under licence from the United Kingdom Home Office.

## Cloning and recombinant fibrinogen expression

Two truncated recombinant fibrinogen α-chain variants (α220 and α390) were produced using Quik-Change II Site-Directed Mutagenesis Kits from Agilent Technologies (Stockport, UK) on the human full-length fibrinogen α610 cDNA and expressed in Chinese hamster ovary (CHO) cells as previously described (*Lord and Binnie, 1993*; *Macrae et al., 2018*). In brief, primers were designed to create stop codons in the cDNA sequence in the expression vector pMLP. Fibrinogen α390 was truncated after aspartate 390, and α220 terminated after serine 220. Truncations were confirmed by sequencing using MRC PPU DNA Sequencing and Services (University of Dundee, UK). The truncated plasmids were co-transfected into CHO cells already expressing the human fibrinogen β- and γ-chains, with a

second plasmid used for selection (pMSV-his). Expression and purification of recombinant fibrinogen was performed as previously described (*Takebe et al., 1995*; *Macrae et al., 2018*). Recombinant homodimer fibrinogen γ′ was produced in CHO-αβ cells transfected with pMLP vector containing mRNA of fibrinogen γ′, expressed and purified as described above. For α390 and α220 fibrinogens, the α-chain size was reduced from 66 kDa to 42 kDa and 25 kDa, respectively. No changes were made to the β- and γ-chains in these variants. The assembled fibrinogen was reduced to 290 kDa for α390 and 256 kDa for α220, from 340 kDa for wild-type (WT). CHO expressing WT fibrinogen, expressing α- and β-chain (CHO-αβ) and β- and γ-chain (CHO-βγ) were kindly provided by Dr Susan Lord. Several recombinant batches of both the truncations and WT were used for experiments. Cells were negative for mycoplasma.

### NativePAGE
Using the NativePAGE system from Thermo Fisher Scientific 1 μg of recombinant fibrinogen was added to 1× sample buffer and loaded onto a NativePAGE Novex 3–12% Bis-Tris protein gel. The gel was run at 150 V for 60 min and then 250 V for a further 60 min. The gel was fixed in a solution of 40% methanol and 10% acetic acid and then heated for 45 s, incubated on a shaker for 30 min, then stained with 0.02% R-250 in 30% methanol and 10% acetic acid, heated for 45 s, and incubated for 30 min. It was destained with 8% acetic acid, heated again for 45 s, and left to destain for 2 hr and imaged using a G:Box (Syngene, Cambridge, UK).

### SDS-PAGE
Integrity of the recombinant fibrinogen chains was assessed by sodium dodecyl sulphate polyacrylamide gel electrophoresis (SDS-PAGE) using the NuPAGE system (Thermo Fisher Scientific). Recombinant fibrinogen (2.5 μg) was added to 1× reducing sample buffer and 1× LDS sample buffer and denatured at 95°C for 10 min, then loaded onto 4–12% NuPAGE Bis-Tris gel. The gel was run at 160 V until adequate separation of the three fibrinogen chains, stained with InstantBlue (Abcam, Cambridge, UK), and imaged on a G:Box.

### Fibrin clot formation and external fibrinolysis
Recombinant fibrinogens were diluted in Tris buffered saline (TBS; 50 mM Tris and 100 mM NaCl at pH 7.5) to a final concentration of 0.5 mg/mL (1.47 μM) in triplicate on a 384-well microtitre plate (Greiner Bio-One, Stroud, UK). Clotting was initiated by the addition of 0.1 U/mL thrombin and 5 mM $CaCl_2$ to each well. Clot progression was monitored on a PowerWave microplate spectrophotometer (Bio-Tek, Swindon, UK) every 12 s at wavelength of 340 nm, for 2 hr at 37°C. After 2 hr, a lysis mix composed of Glu-plasminogen and t-PA in TBS (final concentrations of 0.24 μM and 1 nM, respectively) was added to each well and the plate was read as described above for a further 4 hr. The assay was repeated four times and samples were prepared in triplicate.

### Fibrin fibre arrangement and lysis by scanning confocal microscopy
Each recombinant fibrinogen (truncated variants and WT) was labelled with Alexa Fluor 488 using a protein labelling kit (Thermo Fisher Scientific). Recombinant fibrinogens with a final concentration of 0.475 mg/mL (1.4 μM) were prepared in TBS, and 25 μg/mL of the corresponding Alex Fluor 488 labelled fibrinogen was added. An activation mixture of $CaCl_2$ and thrombin at a final concentration of 5 mM and 0.1 U/mL, respectively, was used to initiate clotting. The sample was added to a μ-slide VI (Thistle Scientific, Glasgow, UK) and allowed to form in a humidity chamber for 1 hr before imaging. Clots were imaged on a LSM880 inverted laser scanning confocal microscope (Zeiss, Cambridge, UK) using a ×40 magnification oil objective, and z-stacks were taken of the middle of the clot (29 slices every 0.7 μm over 20.3 μm) in random areas of the slide. Images were analysed in ImageJ (National Institutes of Health, Bethesda, MD) using an in-house macro to determine fibre count. Clots were prepared on three separate occasions. Fibrinolysis was performed on a clot formed as described above in the μ-slide VI where a lysis mix (plasminogen 0.4 μM and t-PA 6 nM) was added to one side of the slide. Clots were incubated for 15 min to allow perfusion of the lysis mix into the fibrin network. The lysis front was found, and its movement was viewed using the same objective and similar z-stack settings as above (×40 magnification oil objective, z-stacks every 0.6 μm over 20.3 μm).

## Clot structure by scanning electron microscopy

Recombinant fibrinogen with a final concentration of 1 mg/mL (2.94 µM) was clotted in a pierced Eppendorf lid with the addition of 1 U/mL thrombin and 5 mM $CaCl_2$ and incubated in a humidity chamber for 2 hr. After incubation, the clots were washed three times with 0.05 M sodium cacodylate buffer (pH 7.4), fixed overnight in 2% glutaraldehyde prepared in 0.05 M sodium cacodylate. Then the next day, clots were washed a further three times with sodium cacodylate. Clots were dehydrated with increasing percentage of acetone (30, 50, 70, 80, 80, 95, and 100%). The clots were finally subjected to critical-point drying and sputter-coating with 5 nm iridium, before imaging on a SU8230 Ultra-High-Resolution Scanning Electron Microscope (Hitachi, Tokyo, Japan) diameter was measured in ImageJ.

## Protofibril analysis by atomic force microscopy

Fibrin protofibrils and oligomers were analysed using AFM. AFM experiments were performed on the surface of freshly cleaved mica treated with 50 µL of 2 mM $NiCl_2$ for 5 min. After 5 min, the surface was rinsed with $ddH_2O$ and dried with nitrogen gas. For initiation of fibrin polymerisation, a final concentration of 0.02 mg/mL (59 nM) fibrinogen was added to a final concentration of 0.05 U/mL thrombin and 2 mM $CaCl_2$. Due to the swiftness of fibrin polymerisation, the samples were diluted so that the polymer formation could be monitored at 10, 20, and 30 min. After the desired time, polymerisation was interrupted by diluting the polymerising solution a further three times with TBS, followed immediately by the addition of 3 µL of the diluted solution to the surface of pre-treated mica for 10 s. 150 µL of deionised water was then placed onto the substrate for 10 s and removed with nitrogen gas. Imaging of polymers was done using a Nanoscope IV Dimension 3100 AFM (Bruker, Santa Barbara, CA) in tapping mode with a scan rate of 0.8 Hz. Measurements were performed in air using silicon cantilevers (TESPA-V2, Bruker) with a typical radius of 7 nm. Five images were taken from each sample, and 3–4 samples were used for each time point. Images were processed using NanoScope Analysis software. Standard flattening of images was performed, and no resolution enhancement was used. Polymer lengths were measured using ImageJ software. Statistical analysis was performed using OriginLab software (OriginLab Corporation, Northampton, MA).

## Cross-linking experiments

Reaction mixtures composed of 5 µg recombinant fibrinogen variants and 5 mM $CaCl_2$ were mixed with 15 µg/mL FXIII in TBS and followed immediately with the addition of 0.1 U/mL thrombin to activate clotting. The reaction was stopped at 0, 2, 15, 30, 60, and 120 min with a stop solution containing 1× NuPAGE reducing sample buffer and 1× NuPAGE LDS sample buffer in TBS and immediately denatured at 95°C for 10 min and incubated on ice until all time points were completed. The samples were run on a 4–12% NuPAGE Bis-Tris gel at 160 V until adequate separation of the bands. Gels were stained with InstantBlue and imaged on a G:Box. The band densitometry was calculated using ImageJ, the α-chains were normalised to the β-chains, and the percentage remaining was calculated for each time point.

## Fibrin microrheology

Fibrin clot viscoelastic properties were assessed by microrheology using magnetic tweezers as previously described (*Allan et al., 2012*; *Baker et al., 2019*). In brief, recombinant fibrin clots were made with 0.5 mg/mL (1.47 µM) fibrinogen, 5 mM $CaCl_2$ either in the presence or absence of 3.6 µg/mL of FXIII. Superparamagnetic beads, 4.5 µm in diameter, were diluted at 1:250 (v:v) (Dynabeads M-450 Epoxy, Invitrogen, Paisley, UK) were added to the mixture before clotting was initiated with 0.1 U/mL thrombin and then quickly transferred to 0.5 mm diameter capillary tubes (CM Scientific, Keighley, UK). Capillary tube ends were sealed with petroleum jelly to prevent dehydration, and clots could fully form overnight at room temperature. Measurements were collected by applying a force to the paramagnetic beads using four electromagnets to produce a magnetic field gradient. The electromagnets were placed around the sample platform of an Olympus IX71 inverted microscope, which was connected to a CCD camera. Capillary tubes were mounted above the ×40 objective of the microscope. Bead displacement was collected through Labview 7.1 software (National Instruments, Newbury, UK). The displacement of 10 random, individual beads throughout the clot was measured per sample, and this was repeated at least three times.

## Clot contraction

Clot contraction was investigated using a method modified from *Aleman et al., 2014*. 1 mL micro test tubes (Alpha Laboratories, Eastleigh, UK) were coated with Sigmacote, washed with $ddH_2O$, and left to dry overnight. The following morning, blood was collected from the inferior *vena cava* of $Fga^{-/-}$ mice into 10% v/v 109 mM trisodium citrate. Whole blood was diluted 1:3 in TBS and reconstituted with a final concentration of 0.5 mg/mL (1.47 µM) recombinant human fibrinogen. Clotting was initiated by adding an activation mix of 1 pM tissue factor (Stago, Reading, UK) and 10 mM $CaCl_2$. Two tubes containing $Fga^{-/-}$ blood were used as controls, one of which had the activation mix added. Subsequently, images were taken every 30 min for 2 hr, before clots were removed from the tubes and weighed. The non-activated and activated samples were collected and used to quantify the RBC and platelet incorporation into the clot. A sample of non-activated reconstituted blood was taken before activation, and the activated sample was taken after 2 hr. Clot contraction was quantified between 1 and 2 hr using ImageJ. Haemoglobin assay was used to assess RBC incorporation, non-activated and activated samples were diluted in $ddH_2O$ and incubated for 30 min to allow haemolysis, and absorbency was read at 415 nm. RBC incorporation in the clot (%) was calculated as follows: ((pre-$OD_{415}$-post-$OD_{415}$)/pre-$OD_{415}$ × 100). To quantify the incorporation of platelets into the retracted clot, equal aliquots of non-activated and activated samples were diluted 1 in 10 in modified Tyrodes buffer (150 mM NaCl, 5 mM HEPES, 0.55 mM $NaH_2PO_4$, 7 mM $NaHCO_3$, 2.7 mM KCl, 0.5 mM $MgCl_2$, 5.6 mM glucose, pH 7.4) and stained with 5 µg/mL anti-CD41-AF700 (BioLegend) for 15 min. This was further diluted (1:100 in PBS), and CD41-positive events were acquired for 2.5 min at a flow rate of 10 µL/min on a CytoFLEX RUO flow cytometer (Beckman Coulter) and relative events/µL compared using CytExpert (2.1; Beckman Coulter). The assay was repeated three times.

## Platelet-fibrinogen interaction

Flow cytometry was used to investigate platelet-fibrinogen interaction. An antibody staining cocktail of 60 µg/mL anti-fibrinogen FITC (Agilent Technologies, Cheadle, UK), 10 µg/mL CD62P-PE, 5 µg/mL CD63-APC, 5 µg/mL CD41-AF700 (BioLegend, London, UK) was added to 15 µg fibrinogen (WT, α390, α220, and γ'/γ') in modified Tyrodes buffer (150 mM NaCl, 5 mM HEPES, 0.55 mM NaH2PO$_4$, 7 mM $NaHCO_3$, 2.7 mM KCl, 0.5 mM $MgCl_2$, 5.6 mM glucose, pH 7.4). Followed by agonists, PAR4 (200 µM) or ADP (10 µM), or no agonist for basal activity. 5 µL of whole blood collected in 10% v/v 109 mM trisodium citrate from $Fga^{-/-}$ mice was added and rapidly mixed. Tubes were incubated for 20 min in the dark, then fixed with 1% paraformaldehyde in phosphate buffered saline. Control tubes were set up with fibrinogen, CD41-AF700, IgG-PE, IgG-APC, PAR4 (200 µM), and EDTA (10 mM). Samples were acquired on a two-laser four-detector CytoFLEX RUO flow cytometer (Beckman, High Wycombe, UK) and platelets were gated based on their characteristic SSC/FFC profile in combination with a CD41-positive gate, and 10,000 positive events were recorded. Analysis was performed using CytExpert (2.1) to extract MFI values and calculate percentage positive cells based on 2% gates on the background fluorescence of isotype or internal controls, and the assay was repeated three times. Fibrinogen γ'/γ' was used as a control as it lacks the C-terminal AGDV residues, known to bind to the platelet integrin $α_{IIb}β_3$ (*Farrell et al., 1992*).

## Rotational thromboelastometry

Whole blood clot formation and firmness were analysed using a ROTEM Delta (Werfen Limited, Warrington, UK). All investigations were performed through activation of the extrinsic pathway using the EXTEM and APTEM tests. Both tests use tissue factor as an agonist with the EXTEM test influenced by extrinsic coagulation factors, platelets, and fibrinogen. In contrast, the APTEM test allows for the investigation of fibrinolysis, which is based on the EXTEM test but with the addition of aprotinin to inhibit fibrinolytic proteins. $Fga^{-/-}$ mice were bled as described for clot contraction, and the collected blood was reconstituted with recombinant fibrinogen at a final concentration of 0.5 mg/mL (1.47 µM). The assays were repeated three times each. The clot contents of the cups were collected afterwards and fixed overnight in 2.5% glutaraldehyde prepared in 0.9% NaCl (VWR International, Lutterworth, UK). The following morning, clots were washed three times with 0.05 M sodium cacodylate (pH 7.4), dehydrated, critical-point dried, and sputter-coated as described in the clot structure by scanning electron microscopy.

## Statistical analysis

Statistical analysis of experimental data was performed with GraphPad Prism v7 (La Jolla, CA, USA) or OriginLab software (OriginLab Corporation, Northampton, MA), with $p < 0.05$ considered as statistically significant. In all data sets, the results were compared to the control, WT, or pre-treated sample. Data was tested for normality before statistical analysis by ANOVA or Kruskal–Wallis test, followed by post-hoc test of Dunnett, Dunn, or Sidak multiple comparison test. The number of experiments and the statistical tests used for each experiment are detailed in the respective figure legends. Error bars shown as ± standard deviation in all experiments except for early fibrin polymers length which are ±interquartile range.

## Acknowledgements

This research was funded by a British Foundation Programme Grant (RG/13/3/30104, renewal RG/18/11/34036). We thank Martin Fuller (Faculty of Biological Sciences, University of Leeds) for preparation of samples for SEM. We would also like to thank John Harrington and Stuart Micklethwaite from Leeds Electron Microscopy and Spectroscopy centre for their support and assistance in this work. We would also like to thank the Faculty of Biological Sciences Bioimaging facility University of Leeds for the use of the Zeiss LSM880 confocal microscope, which was funded by Wellcome Trust (WT104918MA). We would also like to thank Dr Jay L Degen for kindly providing the $Fga^{-/-}$ mice.

## Additional information

### Funding

| Funder | Grant reference number | Author |
|---|---|---|
| British Heart Foundation | RG/13/3/30104 | Helen McPherson<br>Cedric Duval<br>Stephen R Baker<br>Marco M Domingues<br>Victoria C Ridger<br>Simon DA Connell<br>Helen Philippou<br>Ramzi A Ajjan<br>Robert Ariens |
| British Heart Foundation | RG/18/11/34036 | Helen McPherson<br>Cedric Duval<br>Stephen R Baker<br>Victoria C Ridger<br>Simon DA Connell<br>Helen Philippou<br>Ramzi A Ajjan<br>Robert Ariens |

The funders had no role in study design, data collection and interpretation, or the decision to submit the work for publication.

### Author contributions

Helen R McPherson, Conceptualization, Data curation, Formal analysis, Investigation, Methodology, Resources, Validation, Visualization, Writing – original draft, Writing – review and editing; Cedric Duval, Conceptualization, Resources, Software, Writing – review and editing; Stephen R Baker, Conceptualization, Formal analysis, Investigation, Software, Visualization, Writing – review and editing; Matthew S Hindle, Conceptualization, Formal analysis, Investigation, Visualization, Writing – review and editing; Lih T Cheah, Conceptualization, Formal analysis, Investigation, Writing – review and editing; Nathan L Asquith, Marco M Domingues, Methodology, Writing – review and editing; Victoria C Ridger, Simon DA Connell, Helen Philippou, Conceptualization, Funding acquisition, Writing – review and editing; Khalid M Naseem, Conceptualization, Writing – review and editing; Ramzi A Ajjan, Conceptualization, Funding acquisition, Supervision, Writing – original draft, Writing – review and editing; Robert AS

Ariëns, Conceptualization, Funding acquisition, Project administration, Supervision, Writing – original draft, Writing – review and editing

**Author ORCIDs**
Helen R McPherson [iD] http://orcid.org/0000-0002-3519-498X
Stephen R Baker [iD] http://orcid.org/0000-0002-3147-4925
Ramzi A Ajjan [iD] http://orcid.org/0000-0002-1636-3725
Robert AS Ariëns [iD] http://orcid.org/0000-0002-6310-5745

**Ethics**
Procedures were performed according to accepted standards of humane animal care, approved by the ethical review committee at the University of Leeds, and conducted under license (P144DD0D6) from the United Kingdom Home Office.

**Decision letter and Author response**
Decision letter https://doi.org/10.7554/eLife.68761.sa1
Author response https://doi.org/10.7554/eLife.68761.sa2

## Additional files

**Supplementary files**
• Transparent reporting form

**Data availability**
The source data for Figures 1B-F, Figure 2B and D, Figure 3B, Figure 4, Figure 5B, C and D and Figure 6A-C and D-F and Figure 1—figure supplement 1, Figures 4—figure supplement 1 and Figures 5—figure supplement 1 and 2 and Figures 6—figure supplement 1 are made available as separate source data files.

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
