## [Decision Letter]

**Acceptance summary:**

This paper is of interest to a broad audience in the blood coagulation and fibrinolysis field. Previously undescribed roles in a range of blood clot properties are attributed to a region of the clotting protein fibrinogen, using state-of-the-art methodology. The data support the main conclusions of the paper, open new avenues of investigation for understanding clot properties, and have clinical implications.

**Decision letter after peer review:**

Thank you for submitting your article "Fibrinogen αC-subregions critically contribute blood clot fibre growth, mechanical stability and resistance to fibrinolysis" for consideration by *eLife*. Your article has been reviewed by 2 peer reviewers, and the evaluation has been overseen by a Reviewing Editor and Mone Zaidi as the Senior Editor. The reviewers have opted to remain anonymous.

Essential revisions:

Overall the reviewers were positive about the manuscript and the research work included therein.

1) Please include explanations in your rebuttal regarding the specific items addressed by the two reviewers.

2) Please fix the typographical errors as well as modify figures and legends as suggested for improved clarity.

*Reviewer #2 (Recommendations for the authors):*

The study has, in my opinion, convincingly reached its aims and is an important addition to the field of study, as outlined in my public review. Below is a list of suggestions for additional discussion and minor presentation queries and corrections.

1. I recommend an additional short discussion on the limitations of the study. For example, how aspects of the key novel findings concerning the alphaC-region (fiber growth rate, susceptibility to fibrinolysis, clot strength) could be differentially influenced in an in vivo setting compared to in vitro/ex vivo assays used, and therefore warrant a future approach centered on an in vivo model.

2. The authors mention implications for development of novel therapeutics and thrombosis, targeting the alphaC-region. I recommend this is discussed with additional explanations. Both variants presented gave weakened whole blood clots and are susceptible to fibrinolysis. If the therapeutic goal is haemostasis maintenance with thrombosis prevention it is, to me, unclear how targeting alphaC could achieve this. This could be addressed in point 1 above. For example, a short discussion of whether an animal model of thrombosis would enable assessment of the therapeutic potential of alphaC targeting.

3. In Figure 4 - figure supplement 1, if possible, please include time-lapse images for WT and alpha390 lysis, as for alpha220, and include the timing. Also, I expect the images were intended to be representative rather than "Reprehensive" as denoted in the legend.

4. I recommend that Figure 4 - figure supplement 2 is used as Figure 4E. This and Figure 4D are complementary and compared in the text.

5. Molecular weight markers should be included in Figure 1B.

6. The gene name for the Fga mice should be used consistently throughout the paper.

7. Please address the color coding of Figure 5 - figure supplement 1. Red and green are mentioned in the legend, but blue and orange are in my printed figure.

8. In Figure 5 - figure supplement 2, an s can be added to the word platelet in the legend for part B (, in activated platelets.).

*Reviewer #3 (Recommendations for the authors):*

In the present work the authors determined the contribution of the alfa-C domain and the alfa-C connector in polymerisation, clot mechanical strength, and fibrinolysis by using two recombinant fibrinogen variants alfa-390 (without the alfa-C domain) and alfa-220 (without the alfa-C domain and alfa-C connector). For the first time it is found that the alfa-C connector/domain intervenes in the longitudinal protofibril/fibre growth, mechanical and fibrinolytic stability, and the impact of this finding in terms of designing antithrombotic drugs by targeting this region. The techniques employed to investigate the different roles of these two regions were appropriate. The inclusion of fibrinogen and fibrin degradation studies of the fibrinogen variants it would have been interesting, and would confirmed the thromboelastographic, and external fibrinolysis studies, also the inclusion of a control (mouse blood with its own fibrinogen without adding recombinants) for comparative purposes that probably can be added in the Discussion section as limitations or weakness of the present study. Figure 7 should be withdrawn or improved and incorporated in the results with the respective description.

Some questions and modifications are suggested to the present work.

1. Results section, page 4: Impact of alfa C-subregions on Clot Structure, third paragraph:

Authors measure the fibrin density of the LSCM of the three clots: WT, alfa-390 and alfa-220, and report that the fibrin density of WT and alfa-220 was similar. The three pictures looked different, being the fibrin fibres in WT and alfa-390 homogeneously distributed in space, while in alfa-220 clumps of fibres are seen surrounded by big pores. I wonder if it is correct to report fibrin density values in such inhomogeneous structure, clearly WT is less dense than alfa-390. SEM corroborated the LSCM fibrin structure, and again the spatial disposition of fibrin fibres of alfa-220 was less ordered compared to WT and alfa-390, better seen at X5,000.

2. I will remove figure 7, it is not adding information or clarifying concepts, and is rather confusing, unless authors improve the figure and explicitly explain their model in the result section, especially the longitudinal grow and mechanical stabilisation draws. The premature lysis draw is self-evident.

3. Results section, page 5: Clot Mechanics and in vitro Fibrinolysis, first paragraph: it would be advisable to start describing figure 4 supplement- figure 1A and then 1B.

4. Results section, page 5: Clot Mechanics and in vitro Fibrinolysis, third paragraph:

“whereas α390 clots showed higher elastic modulus compared to WT with FXIIIa at 10 Hz (Figure 4D), likely due to increased fibre branching and clot density in this variant", but without FXIII the structure was the same but not crosslinked, something is missed in the interpretation…

5. Results section, page 5: Clot Contraction, first paragraph: the authors did not find differences between WT and alfa-390 both in the kinetics of clot retraction or clot weight, in spite of the differences in G ´values, could be this be attributed to the fact that siliconized tubes are used? It would be interesting in the future do these experiments in no siliconized test tubes.

6. Results section, page 5: Clot Contraction, second paragraph:

"Comparing the supernatant (post-activation) of contracted clots to corresponding pre-activated samples…": pre-activated is here a confusing term, like a treatment is performed before activation, I would suggest non-activated samples or other that do not generate ambiguity (same correction along the manuscript, Figure 5 —figure supplement 1. Clot incorporation of platelets, Supplementary Materials and methods page 24).

7. Results section, page 6, second paragraph: "However, fibrinogen MFI for α220…". Although MFI is already defined in the methodology, it is suggested to write fibrinogen median fluorescence intensity (MFI).

8. Discussion section, page 8, first paragraph, line 11: "…α2-antiplasmin have also been reported for the D-region and the C-terminal sub-domain….": it would be better to precise and add of the alfa-C domain.

9. Include the limitations and/or weaknesses if there are in the Discussion section.

10. page 22, Heading fibrin clot formation and lysis: it would be more appropriate to indicate that is external fibrinolysis and add this term in page 8, first paragraph line 3 "turbidimetric analysis of external fibrinolysis…".

11. Page 27, Figure 5 —figure supplement 1. Clot incorporation of platelets, legend: the colours in the PDF version are blue and orange.

12. Page 4, paragraph 4, a typo in Figure 2c: C in uppercase.

---

## [Author Response]

Essential Revisions (for the authors):Overall the reviewers were positive about the manuscript and the research work included therein.1) Please include explanations in your rebuttal regarding the specific items addressed by the two reviewers.2) Please fix the typographical errors as well as modify figures and legends as suggested for improved clarity.

We thank the reviewers for their positive evaluation and provide below a point-by-point response to the comments raised. We have also corrected typographical errors and modified figures and legends as requested.

Reviewer #2 (Recommendations for the authors):The study has, in my opinion, convincingly reached its aims and is an important addition to the field of study, as outlined in my public review. Below is a list of suggestions for additional discussion and minor presentation queries and corrections.1. I recommend an additional short discussion on the limitations of the study. For example, how aspects of the key novel findings concerning the alphaC-region (fiber growth rate, susceptibility to fibrinolysis, clot strength) could be differentially influenced in an in vivo setting compared to in vitro/ex vivo assays used, and therefore warrant a future approach centered on an in vivo model.

We agree with the reviewer that a short discussion on limitations would be helpful and have added this additional text on page 8-9 “Our study had a number of limitations. Due to the low expression levels of the αC-truncation variants, any methodologies that require large amounts of protein, including for example in vivo thrombosis studies based on injection of the recombinant variants in FGA-/- mice, could not be performed. Limitations were also posed by the extreme weakness of the α220 variant, which could not be analysed in some of our experiments without crosslinking by FXIIIa. Future in vivo studies using mouse models with similar truncations may provide additional information on the role of the αC-region in physiological haemostasis and pathological thrombosis. However, as mentioned above, such models will likely be confounded by the effects of αC-truncations on both protein function and expression level.”

2. The authors mention implications for development of novel therapeutics and thrombosis, targeting the alphaC-region. I recommend this is discussed with additional explanations. Both variants presented gave weakened whole blood clots and are susceptible to fibrinolysis. If the therapeutic goal is haemostasis maintenance with thrombosis prevention it is, to me, unclear how targeting alphaC could achieve this. This could be addressed in point 1 above. For example, a short discussion of whether an animal model of thrombosis would enable assessment of the therapeutic potential of alphaC targeting.

We thank the reviewers for this comment, and have now added additional discussion regarding potential therapeutic targeting of the αC-region. Overall, our work suggests that targeting the αC-domain may be safer than targeting the αC-connector region, as the impact of the latter may be too drastic, making clots too weak from both a mechanical and proteolysis resistance point of view. However, this will need testing in specific models and experiments since our approach deleted the αC-connector together with the αC-domain, and thus we cannot predict the effect of specifically targeting the αC-connector. Another approach may consist of targeting the αC-crosslinking sites for FXIIIa. A recent study from our lab suggested that this may provide a safe way to prevent thrombosis while not affecting embolism (Duval et al. PNAS 2021). Thrombosis studies in an animal model with a deletion in the αC-region may also contribute new information, but these would be complicated by the low expression levels of fibrinogen with αC-truncations. We have now added further discussion on page 9 to reflect these considerations.

3. In Figure 4 - figure supplement 1, if possible, please include time-lapse images for WT and alpha390 lysis, as for alpha220, and include the timing. Also, I expect the images were intended to be representative rather than "Reprehensive" as denoted in the legend.

As suggested the figure has been updated to include time-lapse of WT and α390 and the times for the α220 images have been included. Apologies for the typo, which has now been corrected to “representative”.

4. I recommend that Figure 4 - figure supplement 2 is used as Figure 4E. This and Figure 4D are complementary and compared in the text.

We agree with this suggestion by the reviewer, and have thus included Figure 4 - figure supplement 2 in the main panel as revised figure 4 D.

5. Molecular weight markers should be included in Figure 1B.

As suggested, we have now included the molecular weight marker for the Native PAGE gel. We have also included a figure 1 supplementary figure 1 with a Native PAGE gel comparing recombinant WT to plasma purified WT.

6. The gene name for the Fga mice should be used consistently throughout the paper.

Thank you for highlighting this inconsistency, we have now updated all murine nomenclature and have consistently used one gene name (FGA^-/-^) throughout.

7. Please address the color coding of Figure 5 - figure supplement 1. Red and green are mentioned in the legend, but blue and orange are in my printed figure.

We apologise for this mistake and have corrected the legend accordingly.

8. In Figure 5 - figure supplement 2, an s can be added to the word platelet in the legend for part B (, in activated platelets.).

We agree and have revised the legend accordingly.

Reviewer #3 (Recommendations for the authors):In the present work the authors determined the contribution of the alfa-C domain and the alfa-C connector in polymerisation, clot mechanical strength, and fibrinolysis by using two recombinant fibrinogen variants alfa-390 (without the alfa-C domain) and alfa-220 (without the alfa-C domain and alfa-C connector). For the first time it is found that the alfa-C connector/domain intervenes in the longitudinal protofibril/fibre growth, mechanical and fibrinolytic stability, and the impact of this finding in terms of designing antithrombotic drugs by targeting this region. The techniques employed to investigate the different roles of these two regions were appropriate. The inclusion of fibrinogen and fibrin degradation studies of the fibrinogen variants it would have been interesting, and would confirmed the thromboelastographic, and external fibrinolysis studies, also the inclusion of a control (mouse blood with its own fibrinogen without adding recombinants) for comparative purposes that probably can be added in the Discussion section as limitations or weakness of the present study. Figure 7 should be withdrawn or improved and incorporated in the results with the respective description.

We appreciate the suggestion from the reviewer regarding the comparison of the FGA-/- supplemented with recombinant human WT fibrinogen to a healthy mouse with normal murine fibrinogen levels. We have performed this experiment and found that FGA-/- supplemented with rec WT hFGN behaved similarly in EXTEM and have included it in the manuscript (figure 6 —figure supplement 1). However, there were some notable differences in the FIBTEM response (lower maximum amplitude for plasma from normal mice compared to plasma from FGA-/- mice supplemented with recombinant human fibrinogen), the reasons of which are not entirely clear to us as yet. The difference may be related to differences between human and murine fibrinogen (e.g. difference in g’ splice variation, aC-region sequence, or other modifications) or to differences in the interactions between murine or human fibrinogen with other functional murine protein partners, but we are unsure regarding the exact reason at this moment in time. Because this observation needs further investigation to establish the cause of the difference, we have decided not to include this data in the revised manuscript. Importantly, however, this does not impact on the main findings which are related to the direct comparison of human WT with aC-truncated fibrinogens.

Some questions and modifications are suggested to the present work.1. Results section, page 4: Impact of alfa C-subregions on Clot Structure, third paragraph:Authors measure the fibrin density of the LSCM of the three clots: WT, alfa-390 and alfa-220, and report that the fibrin density of WT and alfa-220 was similar. The three pictures looked different, being the fibrin fibres in WT and alfa-390 homogeneously distributed in space, while in alfa-220 clumps of fibres are seen surrounded by big pores. I wonder if it is correct to report fibrin density values in such inhomogeneous structure, clearly WT is less dense than alfa-390. SEM corroborated the LSCM fibrin structure, and again the spatial disposition of fibrin fibres of alfa-220 was less ordered compared to WT and alfa-390, better seen at X5,000.

We agree with the reviewer that the confocal images were visually different even if fibre numbers were similar. We have opted to rephrase the text in this section to address the reviewer’s comments.

“However, α220 clots were much more heterogeneous and its structure showed clumps of regions with highly branched fibres and stunted length compared to WT, leaving very large pores that were observed throughout. No difference in average fibre count was observed between WT (12.97±0.35 fibre/100µm) and α220 (14.87±2.25 fibre/100µm p=0.3212) (Figure 2B), but count was significantly increased for α390 clots (19.6±1.79 fibre/100µm; p=0.0041). Despite the clear difference in clot structure between α220 and WT, the average fibre count was similar, suggesting that α220 truncation affects clot organisation and fibre growth, but not the overall number of fibres that are initiated within the clot” Page 4, penultimate paragraph.

3. Results section, page 5: Clot Mechanics and in vitro Fibrinolysis, first paragraph: it would be advisable to start describing figure 4 supplement- figure 1A and then 1B.

We have updated figure 4 supplementary data as suggested by the reviewer so that the figure elements are discussed in sequential order in the revised manuscript.

4. Results section, page 5: Clot Mechanics and in vitro Fibrinolysis, third paragraph: …"whereas α390 clots showed higher elastic modulus compared to WT with FXIIIa at 10 Hz (Figure 4D), likely due to increased fibre branching and clot density in this variant", but without FXIII the structure was the same but not crosslinked, something is missed in the interpretation…

We have rephrased the text for clarity as α390 fibrin clots showed a tendency for higher G’ compared to WT in the absence or presence of FXIII. The text has been rephased to:

“In experiments with and without FXIIIa there was a trend for α390 having a higher elastic modulus compared to WT. This observation is likely due to increased fibre branching and clot density in this variant” Page 5 end of bottom paragraph.

5. Results section, page 5: Clot Contraction, first paragraph: the authors did not find differences between WT and alfa-390 both in the kinetics of clot retraction or clot weight, in spite of the differences in G ´values, could be this be attributed to the fact that siliconized tubes are used? It would be interesting in the future do these experiments in no siliconized test tubes.

The use of siliconised tubes is to prevent the clot edges from sticking to the tube walls. Therefore, it would not be possible to study clot contraction in the absence of silicon as the clot edges would stick to the tube walls.

6. Results section, page 5: Clot Contraction, second paragraph: "Comparing the supernatant (post-activation) of contracted clots to corresponding pre-activated samples…": pre-activated is here a confusing term, like a treatment is performed before activation, I would suggest non-activated samples or other that do not generate ambiguity (same correction along the manuscript, Figure 5 —figure supplement 1. Clot incorporation of platelets, Supplementary Materials and methods page 24).

We appreciate that the wording previously used may have been confusing. As suggested by the reviewer we have updated all references to non-activated and activated (instead of pre-activated and post-activated).

7. Results section, page 6, second paragraph: "However, fibrinogen MFI for α220…". Although MFI is already defined in the methodology, it is suggested to write fibrinogen median fluorescence intensity (MFI).

Revised as suggested.

8. Discussion section, page 8, first paragraph, line 11: "…α2-antiplasmin have also been reported for the D-region and the C-terminal sub-domain….": it would be better to precise and add of the alfa-C domain.

We have revised this to “the C-terminal sub-domain of the αC-region”

9. Include the limitations and/or weaknesses if there are in the Discussion section.

We have now included a brief section on limitations in the discussion on page 8-9, also in response to reviewer 2:

“Our study had a number of limitations. Due to the low expression levels of the αC-truncation variants, any methodologies that require large amounts of protein, including for example in vivo thrombosis studies based on injection of the recombinant variants in FGA^-/-^ mice, could not be performed. Limitations were also posed by the extreme weakness of the α220 variant, which could not be analysed in some of our experiments without crosslinking by FXIII. Future in vivo studies using mouse models with similar truncations may provide additional information on the role of the αC-region in physiological haemostasis and pathological thrombosis. However, as mentioned above, such models will likely be confounded by the effects of αC-truncations on both protein function and expression level.”

10. page 22, Heading fibrin clot formation and lysis: it would be more appropriate to indicate that is external fibrinolysis and add this term in page 8, first paragraph line 3 "turbidimetric analysis of external fibrinolysis…".

Agreed, and now revised accordingly to indicate external lysis.

11. Page 27, Figure 5 —figure supplement 1. Clot incorporation of platelets, legend: the colours in the PDF version are blue and orange.

Thank you for spotting this mistake, which has been corrected in the revised legend.

12. Page 4, paragraph 4, a typo in Figure 2c: C in uppercase.

Thank you for spotting this, now corrected in the revised figure.